# Faecal Microbiota Transplantation, Paving the Way to Treat Non-Alcoholic Fatty Liver Disease

**DOI:** 10.3390/ijms24076123

**Published:** 2023-03-24

**Authors:** María Del Barrio, Lucía Lavín, Álvaro Santos-Laso, Maria Teresa Arias-Loste, Aitor Odriozola, Juan Carlos Rodriguez-Duque, Coral Rivas, Paula Iruzubieta, Javier Crespo

**Affiliations:** 1Gastroenterology and Hepatology Department, Clinical and Translational Research in Digestive Diseases, Valdecilla Research Institute (IDIVAL), Marqués de Valdecilla University Hospital, Av. Valdecilla 25, 39008 Santander, Cantabria, Spain; 2Clinical Trial Agency Valdecilla-IDIVAL, Marqués de Valdecilla University Hospital, Av. Valdecilla, 25, 39008 Santander, Cantabria, Spain

**Keywords:** non-alcoholic fatty liver disease, gut microbiota, dysbiosis, faecal microbiota transplantation

## Abstract

Non-alcoholic fatty liver disease (NAFLD) is currently the most prevalent cause of chronic liver disease (CLD). Currently, the only therapeutic recommendation available is a lifestyle change. However, adherence to this approach is often difficult to guarantee. Alteration of the microbiota and an increase in intestinal permeability seem to be key in the development and progression of NAFLD. Therefore, the manipulation of microbiota seems to provide a promising therapeutic strategy. One way to do so is through faecal microbiota transplantation (FMT). Here, we summarize the key aspects of FMT, detail its current indications and highlight the most recent advances in NAFLD.

## 1. Introduction

Non-alcoholic fatty liver disease (NAFLD) is currently the most common cause of chronic liver disease (CLD), with a global estimated prevalence of 25% of adults [1,2]. However, this prevalence varies between countries, due to differences in age, gender, ethnicity, or dietary habits, among other reasons [3]. NAFLD is also increased in certain at-risk populations, such as patients with type 2 diabetes mellitus (T2DM), obesity and metabolic syndrome (MetS). Unfortunately, the global epidemic of NAFLD seems to be increasing unstoppably, given the prevalence of T2DM and obesity, and thus it is expected that in 2030 the worldwide prevalence of non-alcoholic steatohepatitis (NASH), where steatosis is accompanied by inflammation and ballooning, will increase by 15–56% [4]. Classically, NAFLD was defined as the presence of steatosis in >5% of hepatocytes, without significant ongoing or recent alcohol consumption and in absence of other causes of CLD [5]. Recently, a new definition has been proposed in order to better define patients by reflecting the underlying pathophysiology as a metabolic-driven disease and a shift to “positive” diagnostic criteria rather than exclusion criteria. Those authors proposed a change in the nomenclature of NAFLD to metabolic-dysfunction-associated fatty liver disease (MALFD) [6]. NAFLD is a multisystemic disease [7] that not only has consequences for the liver itself (such as progression to cirrhosis, or hepatocarcinoma) but is also associated with an increased risk of cardiovascular disease and extrahepatic cancer [8,9]. NAFLD also represents a very important economic expense [10], as it is becoming a global public health problem [11].

The pathophysiology of NAFLD is complex and not well understood yet. A theory of “multiple hits” has been proposed [12,13]: the first factor would be hepatic lipid accumulation resulting from the absorption of circulating free fatty acids (FFAs), de novo lipogenesis, and dietary fats; and then, progression to NASH, resulting from secondary hits, such as genetic factors, oxidative stress, mitochondrial dysfunction, pro-inflammatory factors, or gut-liver axis and microbiome alteration [14].

The heterogeneity in treatment response is likely due to the many factors that influence the onset and progression of NAFLD. Therefore, this complexity is probably the reason that today there is no approved effective pharmacological therapy for NAFLD patients. Thus, it seems logical to think that, if NAFLD is a multisystemic disease, it will require a multidisciplinary, holistic and personalized approach [15]. Considering the role of microbiota in the development and progression of NAFLD, the potential to intervene in gut microbiota (GM) represents a promising treatment possibility.

## 2. Gut Microbiome, Dysbiosis and Increased Intestinal Permeability Associated with NAFLD

The human GM consists of 100 trillion microorganisms (i.e., bacteria, viruses, parasites and fungi) that colonize the gastrointestinal tract and help to maintain homeostasis. Specifically, 95% of these are *Firmicutes* (gram-positive), *Bacteroidetes* (gram-negative) and *Actinobacteria* (gram-positive) phyla [16]. Importantly, a disturbance of the balance between beneficial and pathogenic bacteria leads to a condition termed “dysbiosis” [17].

NALFD patients in several studies have been reported to have dysbiosis. A recent meta-analysis, which included 15 studies published from 2012 to 2020, showed that NAFLD patients had a greater abundance of *Escherichia*, *Prevotella* and *Streptococcus* and less of *Coprococcus*, *Faecalibacterium* and *Ruminococcus*, but the significant heterogeneity limited the drawing of conclusions [18]. In non-obese NAFLD patients compared with those without NAFLD an alteration in microbiota has also been described, with an enrichment in gram-negative bacteria (85.21% vs. 71.8%) and a reduction of gram-positive bacteria (14.79% vs. 28.2%). Moreover, a decrease in diversity and a change at phylum-level was observed, with a greater abundance of *Bacteroidetes* and less *Firmicutes* [19].

Likewise, children with NAFLD also exhibited alterations in faecal microbiota, revealing an increase in *Bacteroidetes* and *Proteobacteriae* and a reduction in *Firmicutes* accompanied by a lower α-diversity [20].

Boursier J. et al. [21], reported that gut dysbiosis was associated with NAFLD severity, describing that in patients with NASH individuals it was an increase in *Bacteroidetes* (i.e., *Bacteroidia* class) and a decrease in *Firmicutes* (i.e., *Lachnospiraceae*, *Ruminococcaceae*, *Lactobacillaceae* and *Peptostreptococcaceae* families) [22]. Additionally, differences have also been described among lean, overweight and obese NASH patients [21]. Thus, lean NASH patients had a lower abundance of *Faecalibacterium* and *Ruminococcus*, overweight NASH patients’ faecal microbiota was enriched in *Bifidobacterium* and obese individuals had more *Lactobacilli* [23]. In children and adolescents with NASH, an increase has been observed in *Bacteroidetes* (i.e., *Prevotellaceae* family) and *Proteobacteriae* phyla (Most of the *Enterobacteriaceae* sequences belonged to *E. coli*, presenting a 90% operational taxonomic unit [OTU]). On the other hand, a decrease in *Firmicutes* (at the expense of a decline in *Lachanospiraceae* and *Ruminococcaceae* families) and *Actinobacteria* (with a progressive decrease in the abundance of *Bifidobacterium* from healthy to NASH patients) was observed [22].

Moreover, differences in the composition of the microbiota in faeces according to the degree of fibrosis have also been described. In this regard, an increase in *Bacteroides* and *Ruminococcus* and a decrease in *Prevotella* were associated with severe fibrosis stages [19]. In another study, which included 87 NAFLD patients with liver biopsy, mild/moderate fibrosis stage (F0–F2) was associated with a greater abundance of *Firmicutes*, while in advanced fibrosis (F3–4), *Proteobacteria* phyla were predominant [24]. Similarly, faecal microbiota of children with moderate/severe fibrosis were enriched in *Bacteroidetes*, *Proteobacteria* and TM7 [17].

A recent meta-analysis, which included 15 studies published from 2012 to 2020, showed that NAFLD patients had a greater abundance of *Escherichia*, *Prevotella* and *Streptococcus* and less of *Coprococcus*, *Faecalibacterium* and *Ruminococcus*, but the significant heterogeneity limited the drawing of conclusions [18].

As seen before, the findings are quite variable among different studies (Table 1) and, therefore, several things should be taken into consideration before drawing conclusions. First, we note that the term dysbiosis refers, as mentioned above, to an imbalance and a single responsible pathological microorganism should not be sought. Second, it is known that GM is dynamic and changes in response to environmental factors, such as diet. Third, previous studies that analysed GM in the stool have the limitation that they do not provide information about changes in the small bowel, colon and microorganisms from the mucosal layer adjacent to the intestinal epithelium [25]. This altered GM can disrupt the metabolic pathway and its end products. Thus, an increase in the fermentative pathways has been described (with an increase in the transformation of alcohol into acetaldehyde and acetate), along with a decrease in the concentration of lecithin (because of the increase in the transformation of lecithin into trimethylamine), an alteration in the metabolism of amino acids, such as indole, and an increase in secondary bile acids [26,27]. Furthermore, it has been demonstrated that the increase in endogenous alcohol produced by GM is involved in the development of NAFLD: introducing strains of *Klebsiella pneumoniae*, which produced a high dose of ethanol, into mice induced NAFLD [28]. This raises the question of whether reversing dysbiosis could lead to an improvement in NAFLD.

On the other hand, NAFLD patients have an increased intestinal permeability [29]. It has been proven that the disruption of the intestinal epithelial and gut vascular barrier are early and prerequisite events in the development of NASH and that this is related to dysbiotic microbiota [30]. Mouries J. et al. found that not only bacterial products but also the bacteria themselves will be able to reach the liver, identifying bacteria in the liver parenchyma of mice fed with high-fat diet (HFD) and suggesting their free migration. Accordingly, bacteria and their metabolites can reach the liver, through the portal system, and induce inflammatory response, liver injury and fibrosis [31]. In germ-free mice colonized with stool microbes from 2-week-old infants born to obese mothers (Inf-ObMB), it was shown that Inf-ObMB-colonized mice had an increase in histological signs of periportal inflammation, intestinal permeability and accelerated NAFLD progression when exposed to a Western-style diet [32].

## 3. Different Ways of Managing the Microbiota, Focus on Faecal Microbiota Transplantation (FMT)

In this context, there is an increasing interest in ascertaining how the microbiota could be modulated. Figure 1 summarizes the different ways to manipulate GM. The first way to change GM is diet. A vegan diet is high in fermentable fibre which provides growth substances to microbes [33]. Moreover, in a systematic review, an increase in *Bacteroidetes* at the phylum level and a higher abundance of *Prevotella* at the genus level were observed [34]. Processed food is associated with an impairment of the gut barrier and an alteration of GM. Furthermore, a high-salt diet is associated with a reduction of Lactobacillus abundance [33]. Exercise also affects GM composition. In obese and overweight individuals, energy restriction and a Mediterranean diet with physical activity reduced *Firmicutes*, especially *Lachnospiraceae*, after one year of intervention [35]. Moreover, creating a microbiota profile similar to that of healthy children, with a reduction in *Proteobacteria* and *Gammaproteobacteria,* seems to be useful in children with obesity [36].

Antibiotics are also widely known for their ability to change the microbiota. For example, *C. difficile* is a bacteria that causes dysbiosis and consequently an infection in the colon, which is treated with the non-absorbable antibiotic vancomycin [37]. In a recent study of intensive care unit patients who were administered a broad-spectrum antibiotic, a decrease in α-diversity was observed, with significant differences between bacterial phyla and classes in the stool depending on whether carbapenems or another type of antibiotic was administered [38]. Moreover, antibiotics have been suggested to play a role in functional gastrointestinal disorders [39]. In children, it has also been described that an antibiotic used was associated with a reduction in microbiome diversity and richness, specifically, a meta-analysis showed a reduction in alpha-diversity in relation to macrolide (azithromycin) exposure [40].

A prebiotic is a selectively fermented ingredient that can stimulate the composition and/or activity of the GM, so prebiotics can change GM composition by providing an energy source that can only be used by certain microbes. The fructose-based carbohydrates inulin and fructooligosaccharides are the most common prebiotics and selectively induce proliferation of *Bifidobacteria* and *Lactobacilli*. However, it seems unlikely that any prebiotic is completely specific to a particular bacterial species or genus and hence different individuals may show different responses to the same prebiotic [41]. An increase in butyrate levels has also been observed. This may be related to an increase in lactate-utilizing species, such as *Eubacterium hallii,* which leads to a boost of butyrate or propionate production [42]. Another explanation could be that the alteration of the gut environment with a decrease in gut pH promotes butyrate-producing *Firmicutes* [43].

Probiotics contain live microorganisms (or their components), which are similar to the beneficial bacteria which are usually present in the healthy human gastrointestinal tract. Probiotics can be ingested in the form of any food supplement or as a drug, but usually they are derived from food sources, especially fermented dairy products. The most frequently studied species are *Lactobacillus*, *Bifidobacterium*, and *Saccharomyces* [44]. There are multiple favourable effects that probiotics have in the host: the metabolism of nutrients; namely, to improve digestion, regulate proinflammatory and anti-inflammatory cytokines, decrease the alteration of the microbiome, enhance intestinal barrier function, or enhance the immune barrier function [45]. Beneficial effects have been observed in a variety of intestinal diseases, such as in antibiotic associated diarrhoea, inflammatory bowel disease (IBD) or colorectal cancer [46].

Microbial consortia are natural associations of two or more species acting as a community. Thus, based on recent studies, multi-species and synthetic communities would have a greater effect than single strains. Creating these synthetic communities is quite a challenge: to achieve the ideal cocktail. It is essential to identify the microbiota and its function that is directly related to the disease. To do so, it seems that shotgun sequencing metagenomics is superior to 6S rRNA-based phylogenetic profiling. What is more, it is also essential to predict the bacterial interactions [47]. In vitro, it has been shown that propionogenic bacterial consortium was able to reverse the lack of propionate produced by antibiotic-induced microbial dysbiosis [48].

FMT is a treatment intended to restore a patient’s disturbed GM, by transferring minimally manipulated donor stool to the gut of the patient [49]. There are different types of FMT depending on the type of material and the form of administration. Thus, FMT can be fresh or frozen administered through enema, colonoscopy (in the right colon, except in severe colitis, where it can be applied in the left colon), upper gastrointestinal tract (by gastroscopy, nasogastric, nasojejunal or gastrostomy tube) or by oral capsules. In the case of administration by upper gastroscopy or colonoscopy, the recipients should be prepared with bowel lavage by polyethylene [49].

Different FMT donor screening protocols have been published, in order to try to establish guidelines for choosing the ideal candidate. First, little information is available about faecal donor age criteria, but the European Consensus and the UEG working group recommend individuals aged <60 years to avoid the risk of donor comorbidity [50,51]. Secondly, it should be ensured that the donors are healthy people. For that, some exclusion criteria are usually established, mainly including the risk of infectious disease, gastrointestinal comorbidities and factors that can affect the composition of the FM (antimicrobials or probiotic consumption within the preceding 3 months and during the donation period, major immunosuppressive medications, or systemic antineoplastic agents). Other exclusion criteria also considered are: having a systemic autoinflammatory disease, atopic disease, metabolic syndrome, obesity (Body mass index (BMI) > 30), moderate/severe malnutrition, chronic pain syndromes, ongoing pregnancy, previous or scheduled gastrointestinal surgery, or a history of cancer. Some authors also included diabetes, neurological/neurodegenerative disorders, or chronic treatment (≥3 months) with daily use of proton pump inhibitors [52].

Importantly, this first screening for checking inclusion and exclusion criteria can be made through a questionnaire and a medical interview [53]. Additionally, a stool and blood test at the baseline and periodically is performed for all donors, which is important to avoid the transmission of infectious disease [54]. The Food and Drug Administration (FDA) issued a statement emphasizing the importance of testing multi-resistant microorganisms after two cases of invasive infections caused by extended-spectrum beta-lactamase (ESBL)-producing *E. coli* in two immunocompromised adults. For that reason, the FDA proposed donor screening with specific questions addressing risk factors for colonization with multidrug resistant organisms (MDROs), excluding those at a higher risk of colonization with MDROs and, additionally, MDRO testing of donor stool and exclusion of those that test positive [54]. Given the COVID-19 pandemic, certain extra measures have been taken. First, the initial screening should detail typical COVID-19-associated symptoms within the previous 30 days and a history of travel to more affected areas [55]. Second, in endemic countries, an RT-PCR assay should be conducted for all donors. Finally, taking into account the number of asymptomatic carriers, a molecular stool screening for SARS-CoV-2 should be conducted [56].

Therefore, in relation to a rigorous screening process, the recruitment of stool donors is a challenging process. Paramsothy S. et al. [57] evaluated a faecal donor program, showing that only 12 of the initial 116 respondents (10%) were enrolled as study donors. A recent study in Italy [58] identified only 25% of stool donors as suitable at the end of the selection process. In another recent study in China [59], from 2071 candidates evaluated, only 66 participants (3.19%) finally qualified as stool donors. Another fact to consider is that, even if all patients are healthy people, significant differences between donors’ microbial diversity is observed. 

On the other hand, it has been proposed that, instead of using universal donors, it would be more useful to try to use “super-donors”, which are donors whose stools obtain better results after the FMT than the faeces of other donors [60]. Another new concept is the use of “keystone species” which consists of first performing a metagenomic analysis of the patient’s stool in order to know which species are decreased, and then selecting a specific donor in which those species are increased [61].

After the FMT, monitoring of adverse effects should be carried out, although the specific observation period is currently not well-defined, depending on the way of administration, the baseline characteristics of the patient and the underlying diseases [50]. The most common adverse effects described in the short term are diarrhoea, abdominal cramps, abdominal distension/bloating, abdominal pain, fever, flatulence or constipation, with the majority of adverse effects being mild and self-limiting [62].

## 4. Current Indications for FMT

Today, the use of FMT for recurrent *C. difficile* infection (RCDI) is recommended in various guidelines [34,63]. FMT combination for RCDI (by colonoscopy or nasojejunal tube after 4–10 days of vancomycin) achieves clinical resolution in a significantly higher proportion of patients, compared to fidaxomicin (92 vs. 42%) [64]. Comparing FMT by capsule administration versus by colonoscopy procedure, no differences were demonstrated, since both methods achieved the prevention of RCDI after a single treatment in >95% of participants. However, the capsule group had less adverse events and a greater proportion of patients rated their experience as “not at all unpleasant” [65]. FMT can be repeated in those patients with a recurrence of CDI 8 weeks after an initial FMT. Enema is only recommended if other methods are not available [37].

Another potential indication for FMT is IBD. FMT is effective for IBD patients with RCDI, with a success rate close to 90% after one FMT [66]. Moreover, in patients with IBD and RCDI, a clinical remission of 59% and an improvement in disease activity of 24% has also been described [67]. In addition, in case of failure, a second transplant could be considered. Given the dysbiosis present in IBD and its role in pathophysiology, FMT could also play a role in controlling disease activity [68]. In a recent meta-analysis, clinical remission was quite different among studies, with an overall remission rate of 37% [69]. In patients with mild to moderate Ulcerative Colitis (UC), FMT by colonoscopy was as effective as oral glucocorticoids in inducing remission at week 12, but with the advantage of causing fewer adverse effects [70]. FMT by colonoscopy followed by daily oral capsules is safe and well tolerated in patients with UC, but more studies are needed to draw conclusions as a maintenance therapy [71]. On the other hand, in patients with Crohn’s disease in clinical remission with oral glucocorticoids, a single FMT by colonoscopy showed better results regarding maintenance of remission compared to the placebo group, but without reaching statistical significance [72].

In obesity and MetS the possible benefit of FMT has also been studied. In a recent meta-analysis, that included three randomized placebo-controlled trials (RCTs) with 75 obese patients with MetS undergoing FMT by nasojejunal tube, an improvement in dysglycemia was observed in the short term. However, no differences were shown compared to the placebo in the lipid profile or in the BMI. The results regarding the change in the composition of the microbiota were heterogeneous among the included studies [73]. In another recent meta-analysis that included six RCTs with a total of 154 patients with MetS and/or obesity, FMT group had a lower HbA1c and a better lipid profile in a short timeframe (2–6 week). However, there were no differences in fasting glucose, triglycerides, total cholesterol or BMI. It is necessary to consider the great heterogeneity between the studies in the characteristics of the FMT: the form of administration, the times administered and the type of placebo in the control arm [74]. In the FMT-TRIM trial, there was no difference in the primary endpoint between capsules and placebo (insulin sensitivity at 6 weeks) or in most of the secondary endpoints (including BMI). Nevertheless, it seems that in the subgroup of patients with low baseline microbiome diversity, the improvement in some metabolic outcomes may be greater [75].

Other diseases in which the potential usefulness of FMT is being studied include, for example: irritable bowel syndrome, graft-versus-host disease, or autism spectrum disorders [76].

## 5. FMT in Chronic Liver Diseases (CLD) Other Than NAFLD

Several studies have been published in relation to FMT in patients with CLD in recent years. Bajaj et al. [68]. carried out a study in 20 cirrhotic outpatients with recurrent encephalopathy (HE), defined as at least two documented overt HE episodes requiring therapy, and a MELD score <17. They gave 5 days of broad-spectrum antibiotics prior to a single FMT enema. The sample for the enema was obtained from one donor who had stools rich in *Lachnospiraceae* and *Ruminococcacea*, bacteria that had been reduced in patients with HE. Over 5 months, there were no episodes of HE in the FMT group, compared with five patients (50%) who had HE in the standard of care (SOC) arm. Moreover, in the FMT group, there was an increase in stool microbiota diversity and beneficial taxa (i.e., *Lactobacillaceae, Bifidobacteriaceae*, *Lachnospiraceaeae* and *Ruminococcaceae*), significantly reducing the number of hospitalizations and HE episodes in the long term in the FMT arm compared to the SOC arm [69]. Regarding the stool analysis, differences were observed after >12 months of follow up, with an increase in relative abundance of *Burkholderiaceae* and decrease in *Acidaminoccocaceae*, but not in *Lachnospiraceae* and *Ruminococcaceae* as observed in the previous study. The limitations of these two studies are that the use of pre-treatment antibiotics makes it difficult to discern the role of FMT alone and the small number of participants is another limitation.

Studies with microbiota capsules in cirrhotic patients with recurrent HE are also available. The administration of 15 FMT capsules (with 4.125 g stool from a single donor) at once significantly reduced the number of hospitalizations and the number of subsequent episodes of HE: only one patient (1%) in a FMT group had HE (which was attributed to a transjugular intrahepatic portosystemic shunt placement) compared to three patients in a placebo group (one of them having six episodes) after 30 days of follow up [77]. Furthermore, post-FMT duodenal mucosal diversity increased with higher *Ruminococcaceae*, *Bifidobacteriaceae* and lower *Streptococcaceae* and *Veillonellaceae* at day 30. Nevertheless, there was no difference in stool diversity. One of the limitations of this study is that duodenal biopsies were not repeated in the placebo group. FMT can also result in a better immunoinflammatory state, reducing serum interleukin-6 (IL-6) and lipopolysaccharide (LPS)-binding protein, in relation to an increase in beneficial microbial taxa and improved neurological status [78].

Other frequent complications related to cirrhotic patients are infections, especially those caused by multidrug-resistant germs [79]. FMT, either by enemas or by capsules, was associated with a reduction in the abundance of vancomycin, beta-lactamase and rifamycin antibiotic resistance genes. These findings may represent a new therapeutic target for reducing infections, although further studies are necessary [80].

All these studies included cirrhotic patients of various aetiologies (Alcohol, Hepatitis C virus (HCV), NAFLD and other). However, there are also other publications available about people with chronic liver disease of a specific aetiology, which will be mentioned below.

In non-cirrhotic patients with Primary sclerosing cholangitis (PSC) and current IBD, FMT by enema (90 mL from a single donor) was safe and beneficial, since 33% of patients experienced a ≥50% decrease in alkaline phosphatase (ALP). Furthermore, FMT increases bacterial diversity, which may be related to an improvement in ALP levels. Interestingly, overall bile acid profiles did not change, so it raised questions about whether FMT acts by interactions involving other primary metabolites [81]. However, results should be interpreted and generalised with caution given the small number of patients, the heterogeneity of patient characteristics and the lack of a control arm.

In patients with alcohol-related cirrhosis, FMT enema reduced both alcohol consumption (measured by urinary ethyl glucuronide/creatinine) and cravings [82]. This could be related to the increase in microbial diversity and the reduction in the parameters of systemic inflammation. It should be noted that, in this case, an initial study of the composition of the patient’s stool was carried out in order to later use specific donors that had stool rich in *Lachnospiraceae* and *Ruminococcaceae*, which were absent in the studied population.

Finally, in patients with hepatitis B virus (HBV) and positive hepatitis B virus e-antigen (HBeAg) despite antiviral treatment, it has been described that TMF, but not the placebo, gradually reduced the levels of HBeAg after each dose given (in the duodenum every 4 weeks), achieving the clearance in three out of five patients [83]. A subsequent study supports the potential usefulness of multiple FMT in the duodenum of HBV patients by achieving HBeAg clearance in 16.7% (2/12) of patients after six cycles. Moreover, a significantly reduction in DNA levels was observed compared to the placebo arm; however, no significant reduction in transaminases was observed and no patient achieved hepatitis B surface antigen clearance [84].

To our knowledge, there are currently no published studies conducted in humans on FMT in a cohort of patients with autoimmune hepatitis, primary biliary cholangitis, HCV, Wilson’s disease, alpha_1_ antitrypsin deficiency or hemochromatosis. Table 2 summarizes all the previously mentioned studies.

## 6. Possible Treatments for NAFLD Modulating GM

Currently, lifestyle changes based on diet and exercise are the first step of treatment for patients with NAFLD [5,86]. A Mediterranean hypocaloric diet, low in red and processed meat is recommended. In addition, regular physical activity is also prescribed, with a target of 150–300 min of moderate-intensity or 75–150 min of vigorous-intensity aerobic exercise per week [87]. Lifestyle intervention programs achieve significant weight loss associated with an improvement in histological involvement, reducing steatosis and overall NAFLD activity score (NAS) [88]. Specifically, the ideal goal would be to achieve a 10% weight loss, because from this percentage, improvements in not only steatosis and NASH were reported, but also of fibrosis [89]. However, only a small percentage of patients (10%) will be able to reach the 10% of weight loss and many of these patients fail in the long-term maintenance of the treatment, which is key to prevent weight regain with the passage of time [90].

Prebiotics and probiotics can have a beneficial effect in NAFLD patients by modulating GM. Behrouz V. et al., found an improvement in transaminases and triglycerides after 3 months of probiotic and prebiotic therapy, but did not find significant differences in the rest of the lipid profile or in glucose level [91]. Multiprobiotics consisting of *Bifidobacterium*, *Lactobacillus*, *Lactococcus* and *Propionibacterium* genera, can be useful for liver fat content and transaminases improvement, but they do not seem to have any effect on reducing liver stiffness [92]. Other multiprobiotics, in this case containing six different *Lactobacillus* and *Bifidobacterium* species, also failed to improve liver fibrosis. Moreover, no changes in transaminases, lipid profile, glucose level or hepatic steatosis were obtained [93]. On the other hand, synbiotics are preparations that contain one or more species of probiotic and prebiotic ingredients. Synbiotic treatment, with fructo-oligosaccharides plus Bifidobacterium was ineffective in decreasing liver fat content or in improving liver fibrosis [94].

Pinheiro et al. orally administered a consortium of nine human gut commensal strains to rats fed with a high-fat high-glucose/fructose diet (HFGFD), every 24 h for 2 weeks, and compared them with an oral probing vehicle (sterile PBS) group (HFGFD-VEH) and HFGFD-FMT group. They observed an increase in bacterial diversity and a reduction of portal pressure (PP) compared to HFGFD-VEH. However, the body weight lost was less than achieved in HFGFD-FMT group. No treatment group significantly reversed NASH. There was no difference in glucose and insulin levels, HOMA-IR or cholesterol in any group. However, in the stelic animal (mouse) model (STAM™) study, where a consortium was administrated during 4 weeks, an improvement in NAS was observed, consisting in a decrease in steatosis and ballooning, and in liver fibrosis [95].

## 7. FMT and NAFLD

Regarding animal models ,FMT from human donor with severe steatosis, triggered hepatic lipid accumulation and steatosis in mice [96]. Moreover, oral FMT in HFD mice can efficiently reverse the increase in *Bacteroidetes* and the reduction in *Actinobacteria* and *Firmicutes* [97]. In addition, an improvement in metabolic alteration and liver histology can be obtained, since a decrease in body weight, transaminases, intrahepatic lipid accumulation and NAS score of more than two points has been reported. This improvement can be explained by changes in GM that increase butyrate levels and reduce pro-inflammatory factors (i.e., IL-1, IL-6 or TNFα), promoting an anti-inflammatory microenvironment [74]. However, Mitsinikos et al. showed that FMT may not be as advantageous as dietary modifications, given that FMT from mice receiving dietary interventions to rats with NASH did not improve histological activity [98].

On the other hand, HFGFD can produce, along with an increase in steatosis, an increase in PP. FMT from healthy controls to HFGFD rats with NASH achieves a reduction in PP. This change appears to be related to a reduction in intrahepatic vascular resistance, in association with a significant improvement in molecular markers of endothelial dysfunction. Specifically, an increase in protein kinase B and endothelial nitric oxide synthase were observed [99].

To date, only three clinical trials with FMT have been performed in patients with NAFLD (Table 3). Craven et al. [100], showed that FMT from an allogenic donor, transferred by endoscope to distal duodenum of NAFLD patients, reduced small intestinal permeability, which was measured using the lactulose:mannitol urine test. However, there was no significant difference in the hepatic fat fraction measured by resonance or insulin resistance 6 months post-transplant. This may be a consequence of the fact that changes in the microbiome after allogenic FMT are not very lasting. Witjes et al. [101], transferred faeces from four healthy lean vegan donors to NAFLD patients by nasoduodenal tube. After the transplant, there was an improvement in both the biochemical liver profile and in the necro-inflammation score in the liver biopsy. This consists of a decrease in both lobular inflammation and hepatocellular ballooning, but not in steatosis or fibrosis. Regarding the correction of dysbiosis, after FMT, more *Ruminococcus*, *Eubacterium hallii*, *Faecalibacterium* and *Prevotella copri* were observed. Finally, Xue et al. [102], proposed transplanting faeces from healthy donors by colonoscopy followed by three additional enemas over 3 days. A modulation of GM was observed with a decrease in Proteobacteria and an increase in *Bacteroidetes*, *Firmicutes*, *Fusobacteria* and *Actinobacteria* phyla. In addition, significant improvement in hepatic fat attenuation evaluated by FibroScan was observed, although fibrosis stage was not analysed.

In addition to those mentioned, there are other trials under development: NCT03648086 [103], NCT04465032 [104], NCT04594954 [105], NCT02469272 [106], NCT03803540 [107], NCT04371653 [108], NCT02721264 [109] and NCT02496390 [110]. It should be noted that two of the clinical trials are designed with oral capsules of lyophilized faeces. One of them [108], aims to evaluate microbiome diversity and microbiome richness in faecal samples and the number of participants with an increase in flora diversity after FMT, given twice weekly for 12 weeks. On the other hand, the EMOTION study [111] is a randomized, double-blind, multicentre study, that will be carried out in Spain. It will have two treatment arms, comparing the placebo vs. FMT by oral capsules, and it will be performed in patients with NAFLD confirmed by liver biopsy. As a novelty, it will have an initial phase prior to treatment in which patients will be subjected to lifestyle modifications, in order to stratify patients into those who respond and those who do not respond to lifestyle changes (i.e., leading-phase). In addition, multiple FMT will be performed over a longer period, starting with an initial dose of 24 capsules (6 g of lyophilized faeces) and, subsequently, four maintenance doses of 12 capsules (3 g of lyophilized faeces) every 3 months for 12 months. The main objectives will be to evaluate the safety and tolerability of oral FMT in patients with NAFLD during 72 weeks of treatment and to evaluate the efficacy in hepatic improvement at 72 weeks.

Figure 2 schematically explains what FMT consists of and its usefulness in patients with NAFLD, improving intestinal permeability and dysbiosis.

## 8. Limitations of FMT and Future

These studies also have some limitations and some questions without clear answers First, the lack of changes in specific bacterial taxa may demonstrate that microbiome analysis limited to stool specimens does not properly reflect the changes in the small intestine or microorganisms from the mucosal layer adjacent to the intestinal epithelium. Furthermore, genomic approaches by analysing bacterial 16 rRNA genes are not fully useful for detecting low-abundance microbes that might drive the host phenotype. Another point to consider is that the focus is on bacteria, ignoring the contribution that other microorganisms, such as viruses or fungus, can make. In addition, it is known that GM is dynamic and it changes in response to several environmental factors, such as diet, antibiotics or immune responses. On the other hand, there is a very strict inclusion criteria in trials that may not reflect usual practice, including patients who mostly had non-advanced fibrosis. Moreover, there is a lot of heterogenicity between studies in relation to FMT characteristics: selection of the ideal donor, the way to introduce the material, the grams of faecal stool introduced and the number of FMTs that are needed. Apart from that, it is not yet well-known what factors influence the engraftment, but it seems to be related to taxonomic identity, strain abundance and microbial interaction. Moreover, the complexity and heterogeneity of NAFLD also represents an important impediment in ascertaining the real benefit of FMT, because the problem may be that we are not selecting the patients well, and that it would be effective only for some specific subgroups of patients. Additionally, the studies usually have a short period of follow-up for a chronic disease such as NAFLD [60,112,113,114].

From the data presented above, it appears certain that future work will be related to: first, the better identification and selection of candidate patients who will benefit from FMT and second, improving the selection of the transplanted microbiota, considering the influence of the interaction that the bacteria will have with each other.

## 9. Conclusions

NAFLD is an increasingly prevalent disease worldwide associated with the existing obesity pandemic, largely due to changes in diet and lifestyle that have occurred in recent years. These changes can alter the correct balance between the bacterial species that reside in the intestine, producing dysbiosis and an increase in intestinal permeability. There are more and more studies that associate dysbiosis with the development of NAFLD and that have identified the bacterial species and microbial products involved in its development, although further research is still required on this aspect.

Due to the great complexity of this disease, there are still no effective treatments, but it has been described that FMT can have beneficial effects in other diseases; however, the only well-established indication in the current guidelines for FMT is RCDI. FMT, in all forms of administration mentioned above (enemas, instillation in the duodenum or by capsules) was found to be safe and well-tolerated. In addition, FMT has already been tested in animal models of NAFLD, showing improvements in metabolic alterations, although not at the histological level. Thus, modulation of GM through FMT appears to be a promising new therapeutic strategy in NAFLD, which can lead to a change in the paradigm of treatment of the disease. However, there are still few clinical trials carried out in this area, and it must be considered that both the results and the methodology differ among those that have been carried out. For this reason, further research is needed on the effects of FMT on the resolution of NAFLD and especially fibrosis.

## Figures and Tables

**Figure 1 ijms-24-06123-f001:**
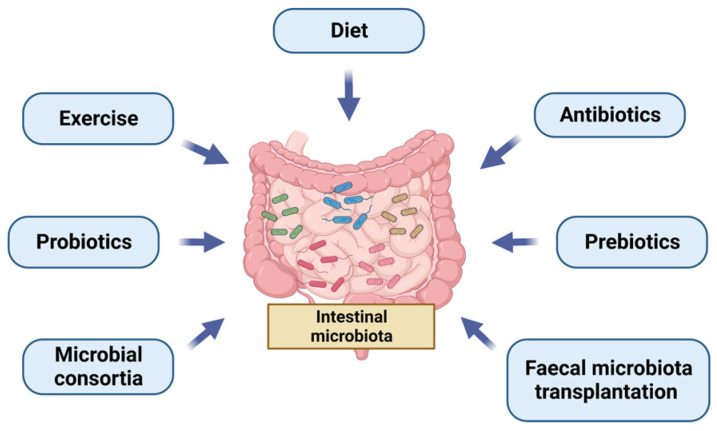
Different ways to manipulate gut microbiota.

**Figure 2 ijms-24-06123-f002:**
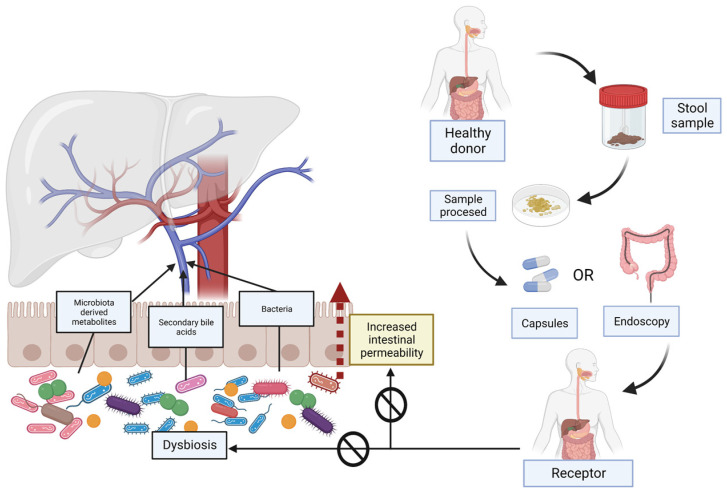
Non-alcoholic-fatty-liver-disease (NAFLD)-related dysbiosis and increased intestinal permeability and the utility of faecal microbiota transplantation (FMT). The increase in intestinal permeability enables bacteria and their metabolites to reach the liver through the portal system. In FMT, a stool sample is taken from a healthy donor. After processing, it will be administered to the receptor subject with NAFLD. The FMT can be performed in different ways, either orally or by endoscopy or enemas. FMT is intended to reverse existing dysbiosis and restore the intestinal barrier, and consequently improve the severity of the disease.

**Table 1 ijms-24-06123-t001:** Summary of studies about dysbiosis observed in patients with non-alcoholic fatty liver disease (NAFLD) and non-alcoholic steatohepatitis (NASH).

Study	Population	Number	Liver Biopsy	Sample	Results
Li F. 2021 [18]	NAFLD patients	15 studies (1265)	9 studies	Stool	Increase in *Escherichia*, *Prevotella* and *Streptococcus* Decrease in *Coprococcus*, *Faecalibacterium* and *Ruminococcus* No differences in *Bifidobacterium*, *Blautia*, *Clostridium*, *Dorea*, *Lactobacillus*, *Parabacteroides* or *Roseburia*
Wang B. 2016 [19]	Non obese adult patients with or without NAFLD	126	-	Stool	Reduction in diversity. Increase in gram negative and decrease in gram positive Increase in *Bacteroidetes* (*Bacteroidia*) Decrease in *Firmicutes* (*Lachnospiraceae*, *Ruminococcaceae*, *Lactobacillaceae* and *Peptostreptococcaceae*)
Schwimmer J. 2019 [20]	Children with NAFLD vs. overweight or obese	124	Yes	Stool	Increase in *Bacteroidetes* and *Proteobacteriae* Decrease in *Firmicutes* and Lower α-diversity; No difference in β-diversity NASH individuals: more *Proteobacteriae* and lower α-diversity F1: increase in *Verrucomicrobia* and *Firmicutes*; F ≥ 2: *Bacteroidetes*, *Proteobacteria* and TM7
Zhu L. 2013 [22]	Children and adolescents with NASH or obesity vs healthy controls	63	Yes	Stool	Low species abundance Increase in *Bacteroidetes* (*Prevotellaceae*) and increase in *Proteobacteriae* (*Enterobacteriae: E: coli*; 90% OTU #20341) Decrease in *Firmicutes* (*Lachanospiraceae* and *Ruminococcaceae*) and decrease in *Actinobacteriae* (*Bifidobacterium*)
Boursier J. 2016 [21]	NAFLD patients	57	Yes	Stool	Increase in *Bacteroides* and decrease in *Prevotella* in NASH patients Increase in *Bacteroides* and *Ruminococcus* in patients with significant fibrosis (F ≥ 2); decrease in *Prevotella* in patients with significant fibrosis (F ≥ 2)
Duarte S. 2018 [23]	Individuals with NASH vs lean healthy controls	23	Yes	Stool	Lean NASH: lower abundance of *Faecalibacterium* and *Ruminococcus* Overweight NASH: enriched in *Bifidobacterium* Obese NASH: enriched in *Lactobacilli*
Loomba R. 2017 [24]	NAFLD patients (F0–2 vs. F3–4)	87	Yes	Stool	F0–2: more abundance of *Firmicutes*; F3–4: more abundance of *Proteobacteria*

**Table 2 ijms-24-06123-t002:** Summary of studies about faecal microbiota transplantation (FMT) in chronic liver diseases (CLD).

Study	Aetiology	Sample Size	Patients Characteristics	FMT Type	Donors	Main Objective	Secondary Aim
Allegretti JR. 2019 [81]	PSC	10	No cirrhosis, 9 UC and 1 CD, only with mesalamine or azathioprine, and 4-week washout period for UDCA	By colonoscopy 90 mL bowel preparation with polyethylene glycol on the day before.	A single healthy donor	Safety: no adverse events related to FMT.	30% experienced a decrease in ALP ≥ 50% during the 24 weeks. Early changes in diversity as from first week that were maintained to 24 weeks. Increase in short-chain fatty acid producing genera. Correlation between the abundance of engrafter OTUs and a decrease in ALP Levels. No changes in stool bile acid profile clustering.
Bajaj J. 2021 [82]	AUD	20	Cirrhosis with a MELD score of 8.9 points	Placebo or FMT enema 1:1 (90 mL, 27 g stool, 2.7 × 10^12^ CFU)	OpenBiome where donor selection was performed to maximize *Lachnospiraceae* and *Ruminococcaceae*, which were lacking in the patients	Safety: 2 patients in FMT group had an adverse event but FMT-unrelated	Reduction of craving in 90%, psychosocial QOL improved and reduction in urinary EtG/creatinine. Reduction in systemic inflammation (IL-6) and in intestinal permeability (lower LBP). Microbial diversity increased with higher *Ruminococcaceae* and other SCFA producing taxa.
Bajaj J. 2017 [61]	Several	20	Cirrhotic with recurrent HE (at least two overt HE episodes requiring therapy), MELD < 17 and no active alcohol abuse.	5 days of antibiotics prior to FMT enema (Three frozen-then-thawed FMT units; 90 mL); Lactulose and rifaximin were continued.	1 donor with the optimal microbiota deficient in HE (*Lachnospiraceae* and *Ruminococcacea*)	Safety: at 150 days, 2 patients (20%) in FMT group had an adverse event but FMT-unrelated.	No FMT patients developed further HE in 5 months follow up vs. 50% in SOC. Improvement in PHES total score and EncephalApp Stroop. Increase in diversity and beneficial taxa (*Lactobacillaceae*, *Bifidobacteriaceae*, Lachnospiraceaeae and *Ruminococcaceae*). MELD score transiently worsened post-antibiotics.
Bajaj J. 2019 [85]	Several	20	Cirrhotic outpatients with recurrent HE	5 days of pre-FMT antibiotics 90 mL enema containing 2.7 × 10^12^ CFU	A single donor: rich in *Lachnospiraceae* and *Ruminococcaceae*	Well-tolerated.	Reduced need for hospitalization and HE episodes. Increase in diversity and increase in relative abundance of *Burkholderiaceae* and decreased *Acidaminoccocaceae* but not in *Lachnospiraceae* and *Ruminococcaceae.*
Bajaj J. 2019 [77]	Several	20	Cirrhotic patients with recurrent HE with MELD < 17.	15 FMT capsules (4.125 g stool) at once vs. placebo No pre-antibiotic therapy	A single donor rich in *Lachnospiraceae* and *Ruminococcaceae*	Safe and well-tolerated.	One patient had an HE (related to TIPS) vs. 3 patients in SOC (1 of them 5 episodes). No differences in stool diversity at day 30. Post-FMT, duodenal mucosal diversity increased with higher *Ruminococcaceae*, *Bifidobacteriaceae* and lower *Streptococcaceae* and *Veillonellaceae.* Reduction in *Veillonellaceae* was seen post-FMT in sigmoid and stool. IL-6 and serum LBP reduced post-FMT.
Bajaj J. 2021 [80]	Several	40 (20 + 20)	Cirrhotic outpatients with recurrent HE	FMT 15 capsules vs. SOC Enema (90 mL) vs. SOC	1 donor rich in *Lachnospiraceae* and *Ruminococcaceae*	Less SAEs in antibiotics + FMT Group.	Beta-lactamase and vancomycin-resistance reduction after FMT, regardless of the mode of administration. No difference in infections.
Bajaj J. 2019 [78]	Several	20	Cirrhotic outpatients with recurrent HE with MELD < 17	FMT capsules vs. placebo	Not specified	Reduction, at 5 months of number of total HE episodes: 6 vs. 1 and in how many patients (3 vs. 1).	An increase in relative abundance of *Lachnospiraceae* and *Ruminococ caceae.* Significant reduction of IL-6. Reduction in total primary BAs and an increase in secondary BAs and secondary/primary BA ratio. No significant changes in MELD.
Chauhan A. 2020 [84]	HBV	29	HBeAg-positive on oral antivirals ≥1 year irrespective of serum levels of HBV-DNA or AST/ALT	In duodenum; 30 g of fresh stool, diluted in 150 mL of saline ×6 cycles at 4 weeks interval	A single healthy donor	Two patients in FMT arm had HBeAg clearance 16.7% vs. 0%.	No achieved HBsAg clearance. DNA became negative faster (25% negative in 6 months). No differences in ALT 6 patients (42.8%) minor adverse events and 1 serious (abdominal pain requiring hospitalization).
Ren YD, 2017 [83]	HBV	18	Persistently positive for HBeAg following >3 years of antiviral; HBV DNA level of <10,000 IU/mL and ALT <80 U/L	FMT to duodenum every 4 weeks until HBeAg clearance was achieved vs. placebo.	Healthy donors.	HBeAg titre declined gradually after each round of FMT;	No HBeAg seroconversion No significant adverse events.

Abbreviations: PSC: Primary sclerosing cholangitis, UC: Ulcerative colitis, CD: Crohn disease, UDCA: ursodeoxycholic acid, FMT: faecal microbiota transplantation, ALP: alkaline phosphatase, OUT: Operational Taxonomic Unit, AUD: Alcohol use disorders, MELD: Model for End-stage Liver Disease, QOL: quality of life, EtG: ethyl glucuronide, CFU: Colony-forming unit, SCFA: short-chain fatty acids, TIPS: Transjugular intrahepatic portosystemic shunt, IL-6: Interleukin 6, LBP: Lypopolysaccharide binding protein, SOC: standard of care, SAE: serious adverse effects, BA: bile acids, HBV: Hepatitis B virus, HBeAg: hepatitis B virus e-antigen , AST: aspartate amino transferase , ALT: alanine amino transferase.

**Table 3 ijms-24-06123-t003:** Summary of published studies on faecal microbiota transplantation performed in patients with non-alcoholic fatty liver disease (NAFLD).

Study	Size	NAFLD Criteria	FMT Type	Main Objective	Secondary Aims
Craven L. 2020 [100]	21 (15 allogenic and 6 autologous)	ASSLD guideline 2018	Endoscope to a distal duodenum (2 g of stool), 3:1 allogenic vs. autologous	No significant decrease in the IR measured by HOMA-IR at six months after FMT	No difference in the hepatic PDFF 6 months post-transplant. Improvement in small intestinal permeability assessed using the lactulose: mannitol urine test. Lower concentrations of non-sterified fatty acids and a decrease in the total:HDL cholesterol ratio. No differences in cholesterol, HDL, LDL cholesterol, triglycerides and APoB:ApoA1.
Witjes J. 2020 [101]	21 (11 autologous and 10 allogenic)	NAFLD by ultrasound	FMT from lean vegan vs. autologous 3 FMT at 8 weeks interval: first by gastroduodenoscopy and then by nasoduodenal tube	Improvement in necro-inflammation score. Trend toward no worsening of fibrosis but not significant. Liver genes: increase in ARHGAP18 and serine dehydratase; and decreased RECQL5 and SF3B3	GGT and ALT decreased. No difference in duodenal microbiota diversity. No significant changes in faecal microbiota diversity, but more *Ruminococcus*, *Eubacterium hallii*, *Faecalibacterium,* and *Prevotella copri.* Change in plasma metabolites: increase in amino acids isoleucine and phenylacetylglutamine.
Xue L. 2022 [102]	75 (FMT 47 vs. non-FMT 28)	ASSLD guideline 2018	Oral probiotics vs. FMT colonoscopy (100 g of faeces with 500 mL of 0.9% saline) + 3 enema (a total of 200 mL of fresh bacteria solution)	Balancing gut microbiota (no statistical differences in Chaol Indexes between two groups after FMT); Fat attenuation degrees decreased from 278.3 to 263.9 dB in FMT group and increased in non-FMT	Decreases in *Proteobacteria* and increase in *Bacteroidetes, Firmicutes, Fusobacteria* and *Actinobacteria.* No differences in blood lipid and liver function.

Abbreviations: ASSLD: American Association for the Study of Liver Diseases; NAFLD: Non-alcoholic fatty liver disease; IR: insulin resistance; PDFF: proton density fat fraction; FMT, faecal microbiota transplant; HDL: high-density lipoprotein; LDL: low-density lipoprotein; APoB: apolipoprotein B; APoA1: apolipoprotein A1; T2DM: type 2 diabetes mellitus; GGT: gamma-glutamyltransferase; ALT: alanine aminotransferase; RECQL5: RecQ Like Helicase 5; ARHGAP18, Rho GTPaseActivating Protein 18; SF3B3, Splicing Factor 3b Subunit.

## Data Availability

Data sharing is not applicable to this article.

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
