# Peer review of "Faecal Microbiota Transplantation, Paving the Way to Treat Non-Alcoholic Fatty Liver Disease"

_ijms, 2023, doi:10.3390/ijms24076123_

Round 1

Reviewer 1 Report

Line 85 ''NAFLD patients have been reported to have dysbiosis and reduced permeability of the intestinal barrier'' , reduced? Are you sure? In general it is increased 

Line 85-105. I recommend an image showing the common findings between the different papers analysed 

Line 220. No reference is made to the microbiota analysis for FMT, currently before the transplant microbiota is analysed, and how? 

Line 244 please use the updated nomenclature for C. difficile

Line 244, I recommend adding more references to other recent reviews showing the potential and limitations of FMT

Line 273 , check the english ''TFM'' 

Line 372 Klebsiella pneumonia please in italics  

Author Response

Response to Reviewer 1 Comments

We are thankful for the thorough review of the manuscript by the Reviewer. In light of the Reviewer’s questions, we have modified the manuscript accordingly (changes underlined in the revised version). We are now submitting an improved version, and hope that the manuscript is considered acceptable for publication.

Line 85 ''NAFLD patients have been reported to have dysbiosis and reduced permeability of the intestinal barrier'' , reduced? Are you sure? In general it is increased

It was indeed a mistake, NAFLD patients have an increased in intestinal permeability.

Line 85-105. I recommend an image showing the common findings between the different papers analysed

We highly appreciate your proposal, in fact a little further down in the test, in table 1 there is a summary of studies about dysbiosis observed in patients with NAFLD and NASH, and where difference and common findings between studies are shown.

Line 220. No reference is made to the microbiota analysis for FMT, currently before the transplant microbiota is analysed, and how?

Thank you for the proposal, we consider very interesting and enriching, so we have added a short paragraph explaining the possibility of “super-donors” and how metagenomic is used for donor selection.

“On the other hand, it has been proposed that instead of using universal donors it would be more useful to try to use “super-donors”, which are donors whose stool get successfully FMT outcomes than the stool of other donors [61]. Another new concept is the use of “keystone species” which consists in first performing a metagenomic analysis of the patient's stool in order to know which species are decreased and, then selecting a specific donor in which those species are increased [62]”.

Line 244 please use the updated nomenclature for C. difficile

Line 244, I recommend adding more references to other recent reviews showing the potential and limitations of FMT

Done.

Line 273 , check the english ''TFM''

Done.

Line 372 Klebsiella pneumonia please in italics

Done.

On the other hand, we have sent the English to be corrected by a native person, hoping that it is more comprehensive and there is no grammatical/spelling mistake.

Reviewer 2 Report

Dear Editor, thank you for giving me the opportunity to peer review this manuscript.

I have found it extremely interesting to read and believe has potential be improved.

Here  below are some suggestions, my biggest concern is that the text is overall confusing and the goal of the review is not always clear. 

Quality of English language must be improved.

1) R 34, 35,46 repetition of "nowadays", R60,63 repetition of "not only"

1)R 48: these are not the only causes of difference in microbiome composition

3) R71 insert reference at the end of the period

4)R91 Please better explain clinical features of patients described in this trial

5) R302-306 Rephrase and insert citation at the end of the paragraph

6)R306 PSC patient who underwent FTM with benefit where at which stage of cirrhosis?

7) Beeing a review, why would you choose to describe NALFD/NASH/ALD/PSC/HBV and not PBC/AIH or HCV?

8) 391-442 this section is very well written and explained. The reading is smooth and it is easy to understand the differences between the various studies considered. The strengths and limitations of the various studies are also well understood.

I find this review of interest but I also find it unorganized and confused. Clusters are not well organized, description of studies does not follow a clear path and information are not always of ease understanding.

I suggest to divide the first paragraph into smaller paragraphs each one explaining which aspect of the microbiome is going to be considered (ex Role of antibiotics, Microbiome composition in NASH, Microbiome composition in children vs adult...)

The information contained in paragraphs two and three is undoubtedly of great interest. However they are not well organized and the ultimate aim of the review seems to drown. Paragraphs often do not have easy-to-understand connections, ranging from the microbiome in NAFLD to the composition of the microbiome in children with NASH, to the microbiome in children with NAFLD, to that of obese patients without a clear address. I also believe that the articles cited are not sufficiently representative of the literature currently available.

As far as paragraph 3 is concerned, I think the title is not explanatory and I don't find it of scientific relevance with respect to NAFLD.

On which bases articles were chosen to be mentioned in the table? It seems to be un-completed

Author Response

Response to Reviewer 2 Comments

We are thankful for the thorough review of the manuscript by the Reviewer. In light of the Reviewer’s questions, we have modified the manuscript accordingly (changes underlined in the revised version). We are now submitting an improved version, and hope that the manuscript is considered acceptable for publication.

1) R 34, 35,46 repetition of "nowadays", R60,63 repetition of "not only"

Following the Reviewer’s suggestion, we have modified the TEXT. Besides, the manuscript has been reviewed by native English speakers.

2)R 48: these are not the only causes of difference in microbiome composition

We are completely agreed with the Referee’s suggestion, and we have modified the sentence.

3) R71 insert reference at the end of the period

The reference is not at the end of the sentence since the last phrase is our opinion and it is not included in the mentioned article.

4)R91 Please better explain clinical features of patients described in this trial

Following the suggestion of the reviewer, we have added details of the characteristics of the study patients and the results obtained.

5) R302-306 Rephrase and insert citation at the end of the paragraph

Done.

6)R306 PSC patient who underwent FTM with benefit where at which stage of cirrhosis?

Thank you for an interesting question. Although it was mentioned in Table 2, we added to the text that the patients were not cirrhotic and that, in addition, all of them had an associated inflammatory bowel disease.

7) Beeing a review, why would you choose to describe NALFD/NASH/ALD/PSC/HBV and not PBC/AIH or HCV?

To our knowledge, there are currently no published studies conducted in humans on FMT in a cohort of patients with autoimmune hepatitis, primary biliary cholangitis, hepatitis C virus, Wilson's disease, alpha1 antitypsin deficiency neither hemochromatosis. We believe that this clarification is very important and we have included it in the text at the end of the section.

8) 391-442 this section is very well written and explained. The reading is smooth and it is easy to understand the differences between the various studies considered. The strengths and limitations of the various studies are also well understood.

I find this review of interest but I also find it unorganized and confused. Clusters are not well organized, description of studies does not follow a clear path and information are not always of ease understanding.

I suggest to divide the first paragraph into smaller paragraphs each one explaining which aspect of the microbiome is going to be considered (ex Role of antibiotics, Microbiome composition in NASH, Microbiome composition in children vs adult...)

The information contained in paragraphs two and three is undoubtedly of great interest. However they are not well organized and the ultimate aim of the review seems to drown. Paragraphs often do not have easy-to-understand connections, ranging from the microbiome in NAFLD to the composition of the microbiome in children with NASH, to the microbiome in children with NAFLD, to that of obese patients without a clear address. I also believe that the articles cited are not sufficiently representative of the literature currently available.

As far as paragraph 3 is concerned, I think the title is not explanatory and I don't find it of scientific relevance with respect to NAFLD.

On which bases articles were chosen to be mentioned in the table? It seems to be un-completed

We are completely agreed with the Referee’s suggestion, and we have made the pertinent modifications. The mentioned section has been organized into different paragraphs depending on whether the patients studied were in NAFLD patients, NASH patients or it was comparing different stages of fibrosis. In addition, we have included a small mention in each section about studies carried out in lean patients and children. Table 2 has also ordered the studies based on what has been mentioned.

Reviewer 3 Report

Comments to the Authors

In Barrio et al, the authors review the roll of microbiota modifications in treatment of chronic liver disease (CLD) primarily non-alcoholic fatty liver disease (NAFLD). They focus on faecal microbiota transplantation (FMT) as a novel approach for the correction of disbiosis connected with NAFLD.  FMT as a novel therapy option for NAFLD is interesting and innovative. Although they are some review papers dealing with gut microbiota and NAFLD this specific subject has not been approached in a review paper as far as I am aware.

The corresponding author has not published original research papers on the topic of FMT but he published the relevant papers about NAFLD and it seems that he is the expert in the field.

The paper has a logical direction, it is written in concise way. In order to be considered for publication, the authors would need to address several minor issues that are described below:

1.       Abstract

-In general, the authors need to avoid repeating phrase (e.g. nowadays, manipulation).

2.       Introduction

-In Introduction section information about NAFLD prevalence and its connections to T2DM and obesity are given, but authors should clarified more closely what NASH is, in term of patohistology and the possibility of progression of NAFLD to NASH and liver fibrosis due to frequent mention of this terms in the text.

3.       Gut microbiome, dysbiosis and increased intestinal permeability associated to 78

NAFLD

-Line 85: instead of reduced permeability should stand increased permeability

-Line 105: instead of NAHS should be NASH

4.       Different ways of managing the microbiota, focus on Faecal microbiota transplantation (FMT)

-Line 190: this repeating phrase “improved the intestinal permeability” usually mean increase of the intestinal permeability and should be replaced with “enhanced intestinal barrier function” or “ decrease intestinal permeability” to avoid misunderstanding.

-Line 225. In this part is significant to emphasize the importance of faecal testing for multi drag resistant organisms, especially bacteria such as E coli (reference:US FDA. Important safety alert regarding use of fecal microbiota for transplantation and risk of serious adverse reactions due to transmission of multi-drug resistant organisms. 2019. https://www. fda.gov/vaccines-blood-biologics/safety-availability-biologics/ important-safety-alert-regarding-use-fecal-microbiotatransplantation-and-risk-serious-adverse)

Also in this section paragraph about contraindications of FMT (such as severe immunodeficiency) and adverse effect of procedure should be added.

5.       Current indications for FMT

-According to meta-analysis reported by Proenca important indication for FMT is obesity and metabolic syndrome and those results should be included in the review (Proença IM, Allegretti JR, Bernardo WM, de Moura DT, Neto AM, Matsubayashi CO, Flor MM, Kotinda AP, de Moura EG. Fecal microbiota transplantation improves metabolic syndrome parameters: systematic review with meta-analysis based on randomized clinical trials. Nutrition Research. 2020 Nov 1;83:1-4.)

6.       FMT in chronic liver diseases (CLD) other than NAFLD

-Abbrevation for hepatic encephalopathy EH should be replace with HE which is used more frequently.

First paragraph in this section needs to be rewrite. I do not understand what this sentence is trying to say, but seems to be important. “They gave 5 days of broad-spectrum antibiotics prior to a single FMT enema, which contained three frozen-then-thawed FMT units (27 grams of stool, 90ml total) of 276 stool from 1 donor with the optimal microbiota deficient in EH (rich in Lachnospiraceae and Ruminococcacea).” Is that donors didn’t have hepatic enepahalopaty?

Limitation of this study is the small number of participants.

-Line 307: Autors should emphasise that this patients were with IBD and PSC, presence of this additional condition could affect the results.

-Line 321: instead of TMF should be FMT

7.       Possible treatments for NAFLD modulating GM

-Line 359:explain the abrevation PP (portal pressure)

8.       FMT and NAFLD

-Line 386: instead of TMF is FMT

-Line 387: Refference should be added at the end of this section: „This change appears to be related to a reduction of intra- hepatic vascular resistance, in association with a significant improvement of molecular markers of endothelial dysfunction. Specifically, an increase of protein kinase B and endothelial nitric oxide synthase was observed.“

-Line 391: authos should refer the question of diferences in FMT succes between lean and obese NAFLD

-Line 393: instead of immpruved permeability should be reduced permeability

-Line 437: Figure 2. Title should be written as a statement instead of “In NAFLD there is a dysbiosis and increased of intestinal permeability.” should be e.g NAFLD related dysbiosis and increased intestinal permeability

Round 2

Reviewer 2 Report

Dear authors,

I have found very interesting changes you have made, specially in table 2. English in now fluent and the paper is well organized. Here just a few suggestions:

table R 369: repetition of also

R 389/390:I would be more precise if you could specify all the etiologies mentioned in the studies instead of " and others"